



**High-resolution Carbon cycling data from 2019 to 2021 measured at six**
**Austrian LTER sites**
Thomas Dirnböck[1]*, Michael Bahn[2], Eugenio Diaz-Pines[3], Ika Djukic[1], Michael Englisch[4], Karl Gartner[4],
Günther Gollobich[4], Armin Hofbauer[4], Johannes Ingrisch[2], Barbara Kitzler[4], Karl Knaebel[1], Johannes
Kobler[1], Andreas Maier[5], Christoph Wohner[1], Ivo Offenthaler[1], Johannes Peterseil[1], Gisela Pröll[1],
Sarah Venier[1], Sophie Zechmeister[3], Anita Zolles[4], Stephan Glatzel[5]
[1] Environment Agency Austria, Spittelauer Lände 5, A-1090 Vienna, Austria
[2] Department of Ecology, Universität Innsbruck, Innsbruck, Austria; Innrain 52, 6020 Innsbruck
[3] Institute of Soil Research, Department of Forest- and Soil Sciences, University of Natural Resources
and Life Sciences, Vienna. Peter-Jordan-Straße 82, 1190 Vienna, Austria
[4] Austrian Research Centre for Forests, Seckendorff-Gudent Weg 8, A-1131 Vienna, Austria
[5] Department of Geography and Regional Research, Faculty of Earth Sciences, Geography and
Astronomy, University of Vienna, Josef-Holaubek-Platz 2, 1090 Vienna, Austria
*corresponding author: Thomas Dirnböck; thomas.dirnboeck@umweltbundesamt.at
Abstract
Seven long-term observation sites have been established in six regions across Austria, covering major
ecosystem types such as forests, grasslands and wetlands across a wide bioclimatic range. The
purpose of these observations is to measure key ecosystem parameters serving as baselines for
assessing the impacts of extreme climate events on the carbon cycle. The data sets collected include
meteorological variables, soil microclimate, $CO_2$ fluxes and tree stem growth, all recorded at high
temporal resolution between 2019 and 2021 (including one year of average climate conditions and
two comparatively dry years). The DOIs of the dataset can be found in the data availability chapter.
The sites will be integrated into the European Research Infrastructure for Integrated European Long-
Term Ecosystem, Critical Zone, and Socio-Ecological Research (eLTER RI). Subsequently, new data
covering the variables presented here will be continuously available through its data integration
portal. This step will allow the data to reach its full potential for research on drought-related
ecosystem carbon cycling.

1. Introduction
Climate change has been affecting ecosystems globally with strong implications for the terrestrial
carbon cycle, which in turn feeds back to the climate system (Heimann and Reichstein, 2008). As an
emerging feature of climate change, extreme climatic events (ECEs) are expected to occur with
increasing frequency and intensity in the coming decades (IPCC, 2021). ECEs are considered to exert
stronger impacts on ecosystems and the services they provide to mankind than gradual changes in
climate (Frank et al., 2015; Reichstein et al., 2013; Grünzweig et al., 2022; Anderegg et al., 2020).
Understanding, predicting and managing extreme climate events and their consequences for
ecosystems and societies will therefore be one of the big challenges in the coming decades. To detect





and attribute impacts of ECEs on ecosystem processes and services they need to be evaluated on the
background of the typical interannual range of these processes (Ciais et al., 2005; Bernal et al., 2012;
Fu et al., 2020; Schindlbacher et al., 2012) and analyses of ecosystem resilience to ECEs require a
robust quantification of baselines of ecosystem functioning (Bahn and Ingrisch, 2018; Ingrisch and
Bahn, 2018). For deriving such baselines as well as interannual variability of ecosystem carbon cycling
coordinated and representative observation networks need to be in place to enable data retrieval as
well as rapid-response scientific campaigns to study after-effects and post-disturbance trajectories
resulting from ECEs (Kulmala, 2018; Mahecha et al., 2017; Mirtl et al., 2018; Dirnböck et al., 2019;
Müller and Bahn, 2022). Datasets obtained through such observation networks are also essential for
benchmarking models (Futter et al., 2023; Baatz et al., 2021; Wu et al., 2018) and for comparison
with ecosystem experiments (Kröel-Dulay et al., 2022).
Within a research infrastructure project focusing on ecosystem carbon, nitrogen, and water fluxes
(LTER-CWN, https://www.lter-austria.at/cwn/), we equipped seven long-term observation sites in six
regions, which are part of the existing Long-Term Ecological Research Network of Austria (LTER), with
high temporal resolution (30-60 minutes) C cycle measurements. The sites cover three major
ecosystem types occurring across Austria (forests, managed mountain grassland, wetlands) and most
of them are part of socio-ecological research platforms for transdisciplinary studies (Figure 1). Here,
we provide observational ecosystem response data capturing naturally-occurring ECEs from the first
three years after the onset of the infrastructure, 2019 to 2021. These data sets include
meteorological variables, soil microclimate, $CO_2$ flux measurements using automated chambers (soil
$CO_2$ efflux) and eddy covariance techniques (net ecosystem exchange), respectively and tree stem
radial increments and shrinkage in forested plots.

## 2.  Site descriptions
The sites are key research infrastructures for ecosystem-related greenhouse gas observations in
Austria. They include forests (Klausen-Leopoldsdorf and Rosalia in Lower Austria, Zöbelboden in
Upper Austria, and Kaserstattalm in Tyrol), mountain grassland (Kaserstattalm, Tyrol), and wetlands
(Pürgschachen Moor, Styria and Lake Neusiedl reed belt, Burgenland). This network of sites covers
typical forest, alpine and wetland ecosystems of Central Europe (Figure 1). Furthermore, the sites
represent different geological characteristics, from crystalline rock in the central Alps to the
limestone in the northern Alps to unconsolidated Holocene sediments in lowlands. All sites are part
of the Austrian LTER network and, once officially launched, will be included in the European eLTER
research infrastructure. For a detailed description of the sites, we refer to the site metadata
catalogue DEIMS-SDR (Table 1).
### 2.1.  Rosalia Forest Demonstration Centre (Mixed beech forest)
The Rosalia Forest Demonstration Centre was settled in 1972, as a cooperation between BOKU and
the Austrian Federal Forests, and has approximately 1000 ha in the western slopes of the Rosalia
Mountains (Rosaliengebirge) in Lower Austria (Figure 1, Table 1). The forest hosts all major tree
species occurring in Austria, i.e. European beech (*Fagus sylvatica* L.), Norway spruce (*Picea abies* (L.)
H.Karst.), Scots pine (*Pinus sylvestris* L.), Larch (*Larix decidua* Mill.), and Fir (*Abies alba* Mill.). The
altitude ranges from 320 to 725 m a.s.l., and mean annual temperature and mean annual
precipitation are 6.5 °C and 796 mm, respectively. Substrate is mainly composed by crystalline rocks,
and soils are predominantly cambisols (FAO, WRB); sporadically in combination with planosols (in
plains and moderate slopes), with fluvisols (in valleys) or podzolic cambisols (steep slopes) (Fürst et
al., 2021).



The demonstration forest holds several experimental and observation sites distributed along its area,
including water, soil, vegetation and air observations (e.g. Gillespie et al., 2023). A watershed (220
ha) is subject to hydrological observations (Fürst et al., 2021), and the forest is regularly monitored
on permanent plots (Gollob et al., 2020). The meteorological data presented here originates from
three stations located at 385 (Mehlbeerleiten), 500 (Kuhwald) and 640 m a.s.l. (Heuberg). The C cycle
data was measured on the DRAIN site, a long-term experiment launched in 2012. The site is located
in a pure mature beech stand at 600 m a.s.l. (47° 42' 26" N; 16° 17' 59" E). It faces north-west, with a
slope of approximately 20 %. The DRAIN experiment focuses on investigating the effect of changing
precipitation patterns for selected soil biogeochemical and microbiological processes (Leitner et al.,
2017; Liu et al., 2019; Schwen et al., 2015; Gillespie et al., 2024). Monitoring is performed on control
and on manipulated plots. The data from both natural and manipulated plots is published with this
paper. Manipulation involves the use of rain-out-shelters (for simulating drought periods of different
length) and of an irrigation system (for recreating rainfall events of different intensity). The
monitoring infrastructure involves the measurements of greenhouse gases (GHG) ($N_2O$, $CH_4$ and $CO_2$)
fluxes, soil nutrients (suction cups) and microclimate parameters.
2.2. Klausen-Leopoldsdorf (Beech forest)
The site, Klausen-Leopoldsdorf, is located about 40 km south-west of Vienna on a NNE-facing slope
and was founded in the 1990ies as one of Austria's ICP Forests site (Neumann and Starlinger, 2001).
The site is divided into four different sub-areas within a small catchment: 1) the ICP Forests Level 2
site, 2) a weather station, located 2.7 km from the ICP intensive plots at 398 m a.s.l., 3) a catchment
runoff weir (475 m a.s.l.), and 4) the LTER-CWN measurement plot (520 m a.s.l.), where the C-cycle
data presented here was measured (Figure 1, Table 1). The forest within the measurement plot is a
pure beech (*Fagus sylvatica* L.) stand. The mean annual temperature is 8°C, mean annual
precipitation is 801 mm (2010-2022). The geological substrate is sandstone, the soil type is mainly
stagnic cambisol/dystric cambisol (FAO, WRB). Instruments installed on the LTER-CWN measurement
area include a sap flow and dendrometer measurement system on 10 trees, 12 GHG automated
measurement chambers for $CO_2$ respiration, soil moisture and soil temperature sensors in different
soil depths (5 – 30 cm).
In addition to the data presented here, many other data sets are available. Soil GHG fluxes (manual
sampling) were measured starting in the year 2001 (Kitzler et al., 2006). On the ICP forest Level 2 site
instruments for long-term monitoring (since 1996) such as soil moisture, air temperature and
humidity, soil temperature, soil solution with suction cups, throughfall deposition, litterfall traps,
stemflow, and manual and automatic dendrometers are installed and the data is available under
https://bfw.ac.at/lims/level2.daten or via the ICP Forests Program Centre.
2.3. Lake Neusiedl (reed belt)
The measurement site is located in the eastern reed belt of the lake and as such inside the National
Park Lake Neusiedl - Seewinkel (Figure 1, Table 1). The region (average altitude 120 m.a.s.l.) is
characterized by a (sub)-continental Pannonian climate with a mean annual precipitation of 576 mm
(2013-2022). The reed belt is a dynamic ecosystem consisting of a mosaic of reed stocks (*Phragmites*
*australis* (Cav.) Trin. ex Steud.), sediment and open water areas. Increasing dry periods and thus
successive drying of the reed belt since 2018 have led to an increase in reed stocks within the belt, as
well as an increase in sediment areas and a strong decline in open water areas, according to a 2021
study that investigated the spatial and temporal variations within the reed ecosystem at Lake
Neusiedl (Buchsteiner et al., 2023). Processes driving $CH_4$ emissions from the reed belt have recently
been investigated in detail (Baur et al., 2024).





The data presented here stems from devices permanently installed on site. They include an eddy
covariance tower for $CO_2$, $CH_4$ and water vapor fluxes and relevant accompanying meteorological
parameters as well as soil heat flux, soil moisture, and soil temperature sensors.
**2.4. Pürgschachen Moor (peat bog)**
The Pürgschachen Moor is located on the bottom of the Styrian Enns valley at an altitude of 632 m
a.s.l. (Figure 1, Table 1). It is a pine peat bog with an extent of about 62 ha. Thus, it is the largest (to a
large part) intact valley peat bog in Austria with a closed peat moss cover and a good example of the
formerly widely distributed peatlands of inner-alpine valleys of the European Alps. The mean average
temperature is 8.2 °C and mean annual precipitation is 1233 mm (2013-2022). The typical vegetation
of the peat bog is constituted of three associations of plants *Pino mugo-Sphagnetum magellanici*
(pine peat bog association), *Sphagnetum magellanici* (coloured bog moss association), and *Caricetum*
*limosae* (bog sedge association), depending on the prevailing hydrological site conditions. The
current mean water table depth is about 14 cm below soil surface at the central peat bog area. Peat
decomposition and related $CO_2$ and $CH_4$ fluxes were subject of a series of research studies (Drollinger
et al., 2019; Knierzinger et al., 2020; Müller et al., 2022; Glatzel et al., 2023).
The data presented here stems from devices permanently installed roughly in the center of the peat
bog. They include an eddy covariance tower for $CO_2$, $CH_4$ and water vapor fluxes and relevant
accompanying meteorological parameters as well as soil heat flux, soil moisture, and soil
temperature sensors.
**2.5. Stubai (subalpine hay meadow, Larch and Spruce forest)**
The two sites reported here are part of the LTER Site Stubai (Table 1), which is located in the Stubai
Alps in Tyrol, Austria (Figure 1). Research at the study site was established in 1993. The two
observation plots are a mountain grassland and a subalpine forest at an alpine pasture area called
"Kaserstattalm". The underlying rock is siliceous and calcareous. The average air temperature is
about 3°C and the precipitation approx. 1100 mm. About 35% of the annual precipitation occurs as
snow during winter months.
The grassland site is located at an altitude of 1810 -1850 m a.s.l on a south-east facing slope with an
inclination of ca. 20°. The site is an extensively managed meadow that is harvested once a year in
early August and grazed lightly in late summer. The soil is a dystric cambisol (FAO, WRB). The
vegetation type is a *Trisetetum flavenscensis* and consists of perennials grasses and forbs dominated
by *Agrostis capillaris* L.*, Festuca rubra* L.*, Anthoxanthum odoratum* L.*, Ranunculus montanus* Willd.*,*
*Leontodon hispidus* L.*, Trifolium repens* L. and *T. pretense* L. (Bahn et al., 2009; Schmitt et al., 2010).
The forested observation plot is located close to the tree line at 1960 m a.s.l. on a slope with an
inclination of 20-35°. It is dominated by the two common tree species European larch (*Larix decidua*
Mill.) and Norway spruce (*Picea abies* (L.) H.Karst.). In former years, the plot was a pasture and it was
reforested in the 1980s (Oberleitner et al., 2022).
Both observation plots are equipped with micrometeorological stations, soil environment monitoring
(soil moisture, soil temperature), and soil $CO_2$ devices. At both observation plots, we measured soil
$CO_2$ fluxes with automated chambers during the summer. The forest plot is additionally equipped
with tree dendrometers and tree sapflow sensors. In the grassland, land use and drought related
carbon cycle research was carried out over the last two decades (Fuchslueger et al., 2014; Hasibeder
et al., 2015; Ingrisch et al., 2020; Ingrisch et al., 2018). Research using the forest plot started only
recently (Oberleitner et al., 2022).
**2.6. Zöbelboden (mixed Beech forest)**



LTER Zöbelboden is located in the National Park Kalkalpen in the Northern Limestone Alps, Austria
(Figure 1). The site Zöbelboden was established in 1992 as part of the UNECE Integrated Monitoring
network (ICP IM) covering a 90 ha catchment with an elevation range of 550 to 956 m a.s.l. (Table 1).
The main underlying rock type is Norian dolomite (*Hauptdolomit*), partly overlain by limestone
(*Plattenkalk*). According to long-term meteorological measurements (1993-2022), mean annual air
temperature and precipitation are 8.2 °C and 1645 mm, respectively. Maximum precipitation occurs
in summer and snowfall usually between December and April.
The data presented here was measured at the Intensive Plot II situated on a steep (36° on average)
north-westerly exposed slope at 880 m a.s.l. The soils of the plot are lithic and rendzic leptosols (FAO,
WRB). The plot is dominated by beech (*Fagus sylvatica* L.) with intermixed sycamore (*Acer
pseudoplatanus* L.), European ash (*Fraxinus excelsior* L.) and spruce (*Picea abies* (L.) H.Karst.). Since
the year 1995, this plot is equipped with a number of field measurement devices for long-term
monitoring (throughfall deposition, litter fall traps, lysimeters, soil moisture and temperature
sensors, manual dendrometers) and supplemented by other monitoring activities (tree inventory,
needle and leave chemistry, soil chemistry, etc.; see e.g. Leitner et al., 2020; Kobler et al., 2019;
Dirnböck et al., 2016; Dirnböck et al., 2020). Drought-impacts on carbon allocation in the forests of
the catchment is currently one of the research foci for which long-term observation data exists (see
e.g. Hartl-Meier et al., 2014) as well as experimental plots with rainout shelter.
The instruments and data included here are soil respiration automated chambers, soil water
potential and temperature sensors as well as automated dendrometers. The meteorological data
stems from a station in close proximity at the plateau at 890 m a.s.l.. The site is also equipped with
an Eddy covariance tower, but this data will be published elsewhere.

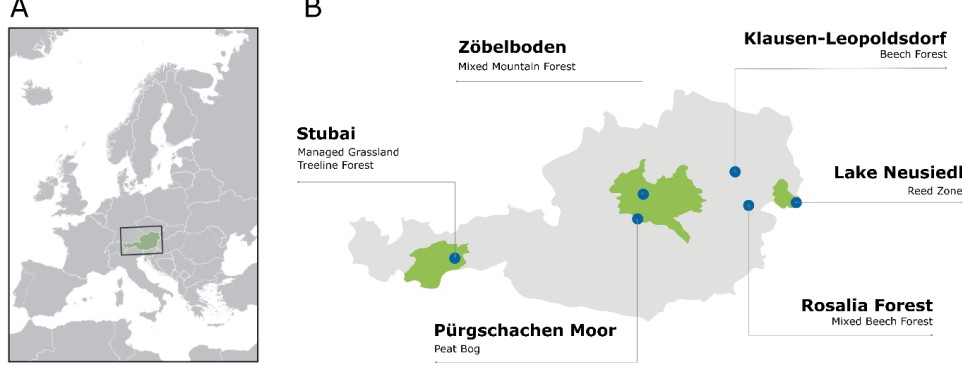


*Figure 1*. Location of sites in A) Europe and B) Austria. Blue dots indicate the sites; green areas are socio-ecological research
platforms (from left to right: LTSER Tyrolian Alps, LTSER Eisenwurzen, and LTSER Lake Neusiedl) within the LTER Austria
network.
*Table 1. Metadata of the sites and observation plots including geographic boundaries, linked data sets, etc. can be found in
the site and dataset registry system DEIMS-SDR.*

| Site | Site Code | Observation plot | DEIMS.iD |
|---|---|---|---|
| **Rosalia Forest Demonstration Centre** | ROS | | https://deims.org/77c127c4-2ebe-453b-b5af-61858ff02e31 |
| | | Heuberg Meteorological Station | https://deims.org/locations/44854b32-64c3-4c9d-9aec-9b0b74f8ac70 |



| | | Kuhwald Meteorological Station | https://deims.org/locations/1225d57e-02da-47fd-9760-ab39d64999ef |
|---|---|---|---|
| | | Mehlbeerleiten Meteorological Station | https://deims.org/locations/0becf0ce-98d7-4f64-a074-f89046083e5e |
| | | DRAIN Station | https://deims.org/locations/b7008603-fca2-452f-9b3d-aad30cdafc7a |
| **Klausen-Leopoldsdorf** | KLL | | https://deims.org/locations/bb472a51-f85f-4de0-8358-f21ecbe2a102 |
| | | Measuring station | https://deims.org/locations/d5cba3ce-7489-46d1-8d97-61641ffb5758 |
| **Lake Neusiedl** | NSS | Same as site | https://deims.org/locations/4234987b-9031-4332-9bdd-f869d503ac51 |
| **Pürgschachen Moor** | PUE | Same as site | https://deims.org/locations/ab2d021b-f318-487a-a85b-ab34566e4c02 |
| **Stubai** | KAS | | https://deims.org/324f92a3-5940-4790-9738-5aa21992511c |
| | | Kaserstattalm meadow | https://deims.org/locations/cf7843b7-32d6-44e9-ba82-9a8d915036a7 |
| | | Kaserstattalm forest | https://deims.org/locations/af2afdad-d6fb-4580-b6e3-be7d07b56f8e |
| **Zöbelboden** | ZOE | | https://deims.org/8eda49e9-1f4e-4f3e-b58e-e0bb25dc32a6 |
| | | Intensive Plot II | https://deims.org/locations/bc96a499-1b20-4da8-be2d-17306d64b788 |


## 3. Dataset description, measuring methods, QA/QC


We followed routine quality assurance (QA) and quality control (QC) procedures to ensure
functionality of the sensors and data quality comprising remote function control, on-site check of
sensors and cables, regular sensor calibration, data checks through different quality assurance
procedures (e.g. exceedance of thresholds, outlier detection, deviations from other measurements),
and data quality flagging.

### 3.1. Meteorology, soil temperature and soil moisture

All meteorological stations are located within the boundaries of the respective sites except for
Klausen-Leopoldsdorf, where the station is at a distance of 2.7 km from the site. Meteorological
measurements in the wetland sites were implemented next to the Eddy Covariance tower. In
addition to the routine data checks, we compared the measurements with nearby stations where
appropriate. Meteorological measurements were detected in a one-minute-interval and averaged
over half-hour periods while rain data was summed. The measurements include air temperature,
precipitation, humidity, wind speed and direction, air pressure, and several radiation variables (at
least global radiation, but also short- and longwave radiation, photosynthetic active radiation, etc.).

We used different types of soil temperature and soil moisture or soil water potential sensors,
respectively (PT100 or thermoelements for soil temperature, TDR or FDR-sensors for soil moisture, and
soil water potential sensors). Before we buried the soil temperature or soil moisture and soil water
potential sensors into the soil, they had been calibrated or at least tested for consistency. Mostly, we
used gravimetric samples to calibrate the TDR and FDR soil moisture sensors. At Zöbelboden, where
stony, organic rich soils occur, we corrected the TDR values using water potential sensor data installed
in the same soil profiles together with soil water retention functions derived from undisturbed soil



cores. In addition to the regular QC procedures, we checked the data for consistency of the values
across sensors (e.g. along the soil profiles) and compared them with other measurements (air
temperature and precipitation).
## 3.2. Carbon fluxes
### 3.2.1. Soil $CO_2$ efflux
We measured soil $CO_2$ efflux at five of the seven observation plots. The automated soil $CO_2$ respiration
measurement systems are capable of operating autonomously during the snow-free periods. The
measurement chambers and measurement systems collected air from the chamber headspace
continuously to determine the exchange of $CO_2$ between soil and atmosphere at the observation plots.
In all sites, we used non-steady state, non through-flow chambers (Pumpanen et al., 2004). In addition
to the automated systems, manual flux measurements were also performed which served to validate
the automated measurement systems. Table 2 provides detailed information on the measurement
systems used at the sites.
Two different automated chamber systems were used: a LI-COR System and custom-made chambers
in combination with LI-COR trace gas analysers (Table 2). The custom-made soil chambers are
equipped with a fan and a thermometer. The controlling unit and the gas analyzer (either a $CH_4/CO_2$
LI-COR 7810, a LI-COR 840, or a LI-COR 8100A, LI-COR Biosciences, USA) are located in already
existing measurement containers. Remote access to the devices allows for checking plausibility of the
data and chamber leakage in real time. We visited the instruments at weekly to monthly intervals,
with maintenance and supervision works including a check of the tightness of the gas lines,
connections and chamber lids, the correct closing and opening of the chambers and the functioning
of ventilation fans inside the chambers, ingrowth of plants, and the gas analyser. The gas analysers
were calibrated once a year in the laboratory with calibration gases. We de-installed and serviced the
chambers during winter but frames stayed permanently on site to avoid disturbance of the soil.
At Klausen-Leopoldsdorf, the gas fluxes of readings were determined using the R package "gasfluxes"
(Fuss, 2020). At Rosalia, a custom-made Python script was used. Zöbelboden and Kaserstattalm
process the data with SoilFlux Pro Software (LI-COR Biosciences, 2019). We used the $R^2$ of the fitted
empirical models to select valid data. We refer to Table 2 and the metadata published with the data
for the detailed specifications.
*Table 2.* Specifications of the different soil CO2 flux systems following the standard of (Bond-Lamberty et al., 2021).

| Field Name | Description | Unit | Klausen-Leopoldsdorf | Stubai grassland | Rosalia | Zöbelboden |
|---|---|---|---|---|---|---|
| System | | | **auto** | **auto** | **auto** | **auto** |
| GHG chambers | | | Custom-made (n=12) | LI-8100-104 (n=4) | Custom-made (n=12) | LI-8100-104 (n=6) |
| INSTRUMENT | Measurement instrument model | | LI-COR LI-7810 | LI-8100A | LI-840 | LI-8100A |
| MSMT_VAR | Type of flux measured | | Soil respiration (Rs) | | | |
| AREA | Soil surface measurement area | cm$^2$ | 2500 | 317.8 | 2500 | 317.8 |





| VOLUME | Volume of measurement chamber | cm³ | 37500 | 4076.1 | 37500 | 4076.1 |
|---|---|---|---|---|---|---|
| V/A | Volume/Area ratio | cm | 15 | 12.83 | 15 | 12.83 |
| COLLAR_DEPTH | Depth of collar insertion | cm | 5 | 2 | 10 | 2 |
| OPAQUE | Opaque chamber | | no | yes | no | yes |
| chamber system | static chamber - closed or open | | non-steady state, non through-flow chambers | | | |
| closing time | closing time of chamber (=time used for flux calculation) | sec | 175 | depending on year | 1620 | 210 |
| PLANTS_REMOVED | Plants removed from inside the collar | | no, but hardly any | yes | no, but hardly any | no plants |
| flow_rate | sample flow rate through tubing | l min⁻¹ | 1 | 1 to 2 | 0.25 | 1.7 |
| FAN | Mixing fan in chamber? | | yes | no | yes | no |
| CRVFIT_CO2 | Flux computation method ("Lin" or "Exp" for linear and exponential, others) | | linear | automated[1] | Lin/HMR[2] | Automated[1] |
| R2_CO2 | $R^2$ of flux computation | fraction | 0.90 | 0.95 | 0.95 | 0.99 |
| Calculation of flux | | | R Package gasfluxes | LI-COR Soilflux Pro | custom-made python script | LI-COR Soilflux Pro |

[1] "Exp" in the data indicates that the exponential fit was better than the linear fit (Exp_SSN<Lin_SSN). "Lin" indicates that the linear fit was better after the maximum number of iterations; the non-linear coefficients have therefore been derived from the linear fit.
[2] Hutchinson and Mosier (1981)

3.2.2. Eddy Covariance measurements at wetland sites

In both wetland sites, the Pürgschachen Moor and Lake Neusiedl, fully equipped Eddy-Covariance
systems are in place. Wind speed and direction were measured using a three-axis ultrasonic
anemometer (WindMaster Pro, Gill Instruments, Lymington, UK). $CO_2$ and $H_2O$ mixing ratios were
measured using the closed-path infrared gas analyser LI-7200 while $CH_4$ was detected with the open
path gas analyser LI-7700 (both LI-COR Inc, Lincoln, USA). The measurements were performed with a
sampling rate of 10 Hz. We installed the devices at a vegetation dependent height, 3.05 m above
ground in the Pürgschachen Moor and in the reed belt of Lake Neusiedl 8.6 m, respectively. The Eddy
Covariance devices were checked daily via remote access, calibrated once a year, and monthly in the
field.



The EC data contains half-hour eddy covariance flux measurements for $CO_2$, $CH_4$ and water vapor. We
calculated the fluxes with the EddyPro® Software package in the Express mode with default settings
(double rotation, block averaging, covariance maximization, etc.) as part of the SmartFlux® 2 System,
providing fully corrected and valid fluxes with quality flags ranging from 0-2. The final flags are based
on a combination of partial flags accounting for steady state and turbulent conditions. Only fluxes
flagged with 0 (best quality fluxes) or 1 (fluxes suitable for general analysis such as annual budgets)
are shown in the data. Gaps in the data-set result from missing micro-meteorological conditions,
from data cleaning due to the quality flags or from power breakdowns.
### 3.3. Radial tree stem growth at forest sites
Zöbelboden, Klausen-Leopoldsdorf, Rosalia used the DR26 sensor (EMS, Brno, Czech Republic), Stubai
used Ecomatic DC2 (Germany) for registering the radial stem increment. Maintenance involved
avoiding any shift of the sensor during the operation. Concerning data quality and control methods the
Mini32 software (EMS, Brno, Czech Republic), includes graphical features to process the measured
stem increment data. Data processing comprises outlier detection by visual assessment based on
expert knowledge. Ecomatic raw data was treated with custom-made R scripts. In both cases,
unrealistic values beyond the slowly increasing linear growth rates were visually assessed and deleted.
## 4. Data file structure
We used the eLTER Data specification, which is available on Zenodo
(www.doi.org/10.5281/zenodo.6373409). Apart from the data files, the measurement locations
(Station files) and the sensors (methods) are included.

## 5. Data validation
In most sites, the year 2020 did resemble an average climate with mean annual temperatures and
precipitation sums close to the long-term averages, whereas either 2019 or 2021 were drier and, in
some cases, also warmer compared to the long-term average (Figure 2). Differences in the seasonal
precipitation patterns between these years vary a lot between sites. In sum, the dry periods resulted
in lower precipitation in 2019 and 2021 in all sites. The mean annual temperature maxima (90
percentile) were between 0.3 °C (KAS) and 2.3 °C (ZOE) higher in 2019 than in 2020. These
differences were lower in 2021 (< 0.5 °C). At KAS, the maximum temperatures in the year 2021 were
lower (0.6 °C). In accordance with precipitation and temperature, soil water content showed the
lowest values during the years 2019 and 2020, and soil temperature were higher during these years
(Figure 3).

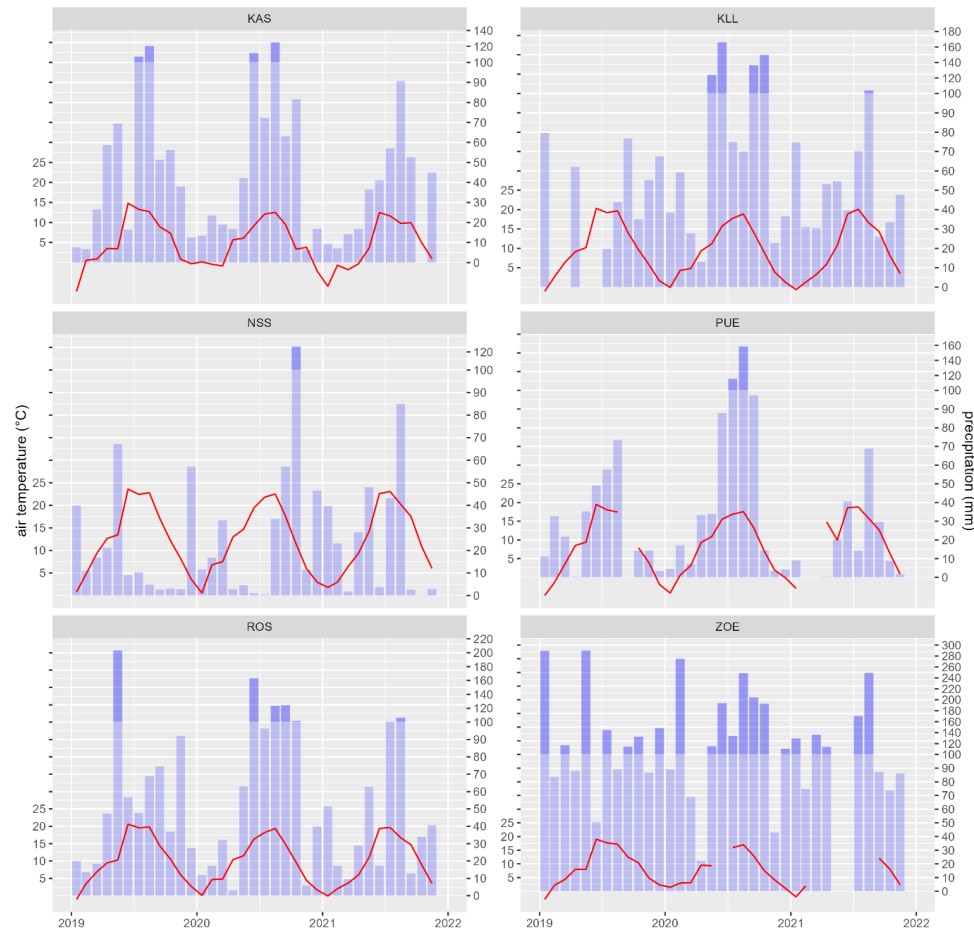


*Figure 2*. Monthly mean air temperature (red line) and monthly precipitation sums (blue bars; different scale > 100 mm) at
the six regions.

We measured soil $CO_2$ respiration at four sites (Figure 4). The complexity of automated chamber
measurements resulted in some data gaps: at KAS and ROS during the years 2019 and 2021
respectively; at KLL and ZOE, the respiration data covers most of the snow-free period.

At Klausen-Leopoldsdorf (KLL) and Zöbelboden (ZOE), we compared the automatically measured soil
$CO_2$ flux rates with manual measurements. For both sites, we used a portable infrared gas analyzer
(EGM-4) connected to a manual soil respiration chamber (SRC) (PP Systems International Inc.,
Amesbury, MA, USA). The two measurement sites were equipped with permanently installed collars
(KLL: randomly distributed within the site in immediate vicinity of the automated chambers (n = 12);
area = 284 cm² and 2 cm insertion depth; Zöbelboden: regular grid covering the entire plot (n = 30),
area = 78 cm² and 1.5 cm insertion depth). The chamber closure time was 60 and 100 seconds in KLL
and ZOE, respectively. Manual measurements took place in monthly intervals from Oct. 2019-Jun.
2020 at Klausen-Leopoldsdorf and from Jun. 2019 until Oct. 2019 (monthly interval) and in July 2020
(diurnal variation) at ZOE. Rs was calculated automatically by fitting a linear (KLL) or quadratic
function (ZOE; quadratic fit for flow rates > 0.2 ppm s$^{-1}$, otherwise a linear fit was used) to the
increasing $CO_2$ headspace concentration.

The mean $CO_2$ fluxes of the automated chambers correlated well with the manually measured fluxes during the measurement campaigns (Figure 3). At KLL, the $R^2$ was 0.95 ($p$-value < 0.05), at ZOE it was 0.85 ($p$-value < 0.05). In both sites, neither the intercept nor the slope was significantly different from 0 ($p$-value > 0.2) and 1 ($p$-value > 0.49), respectively. At ZOE, the spatial flux variation was much higher than at KLL (Figure 3A and 3B). This reflects the heterogeneity of the soil conditions (shallow rendzic leptosols with interspersed fine-scale patches of deeper soils), the canopy gaps (with lower root density), and the uneven distribution of litter due to the steep slope at the plot, more effectively captured in the manual measurement (n=30) than by the automated chambers (n=6). In summary, we conclude that the spatial variation in $CO_2$ fluxes was higher at both sites than the difference in fluxes caused by the measurement devices.

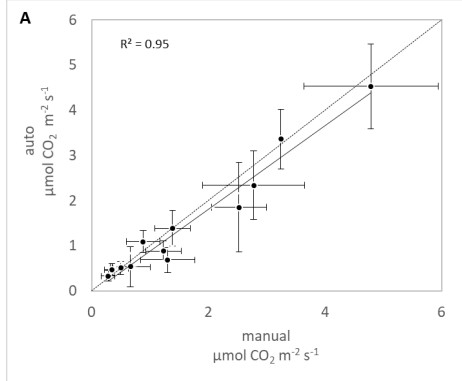

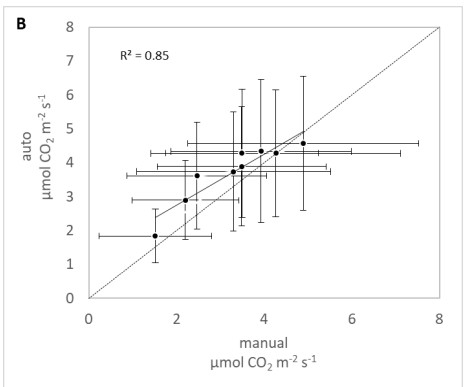

*Figure 3.* Comparison of automated and manual soil $CO_2$ fluxes at A) Klausen-Leopoldsdorf and B) Zöbelboden. See *Table 2* for the specification of automated chamber data. Error bars indicate spatial variation (standard deviations).

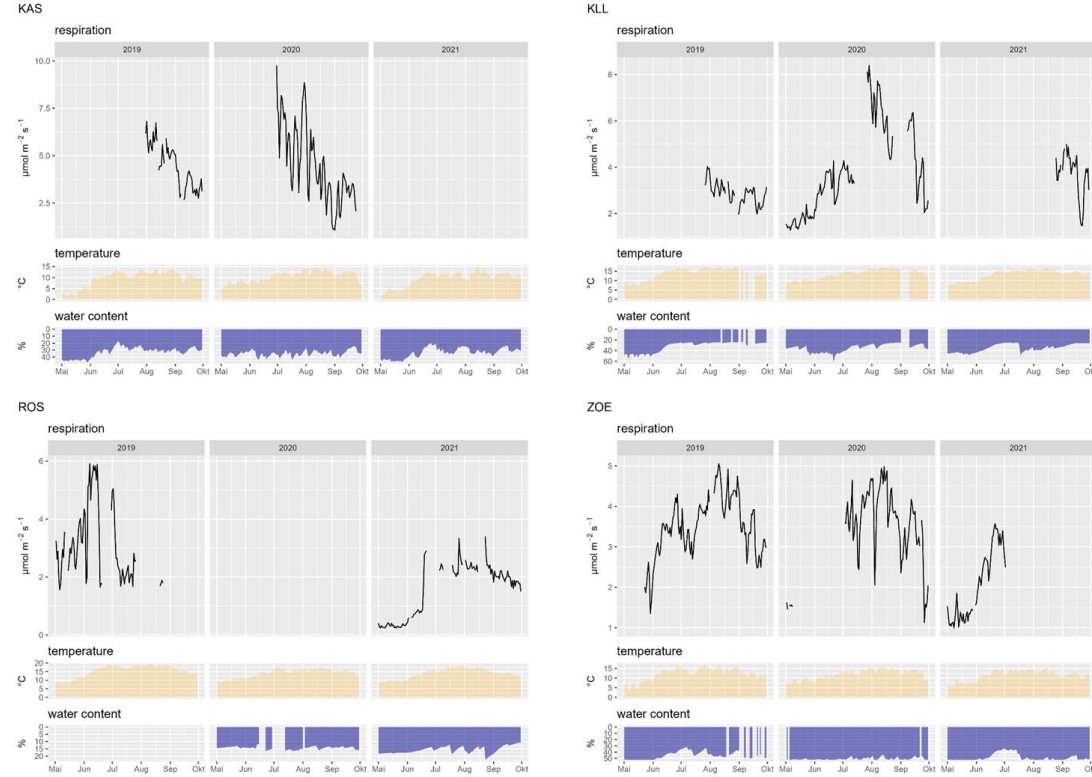


*Figure 4. Soil CO$_2$ respiration (mean of all chambers), soil temperature (mean of sensors in 5-15 cm depth) and soil water*
*content (mean of sensors in 5-15 cm depth) in the forested sites Kaserstattalm forest (KAS), Klausen-Leopoldsdorf (KLL),*
*Rosalia (ROS), and Zöbelboden (ZOE).*

Soil CO$_2$ fluxes are temperature dependent, thus closely follow soil temperature (Figure 4). Their
additional limitation through soil water availability for plant metabolism and microbial activity is
much less pronounced. For an interpretation of the CO$_2$ respiration fluxes and their limiting factors,
we refer to the citations listed in the site description chapter. Drollinger et al. (2019) provides
interpretations of the patterns of CO$_2$ and CH$_4$ fluxes, measured using Eddy covariance techniques at
the bog site Pürgschachen Moor (PUE), and likewise, Baur et al. (2024), for the reed belt of
Neusiedler See (NSS). Stem growth limitations can, on the other hand, be closely related to soil water
content, particularly at sites with relatively low precipitation such as Klausen-Leopoldsdorf (KLL)
(Figure 5). For an in-depth study of drought related effects on tree growth at the treeline forest at
Kasterstattalm (KAS), we refer to Oberleitner et al. (2022).

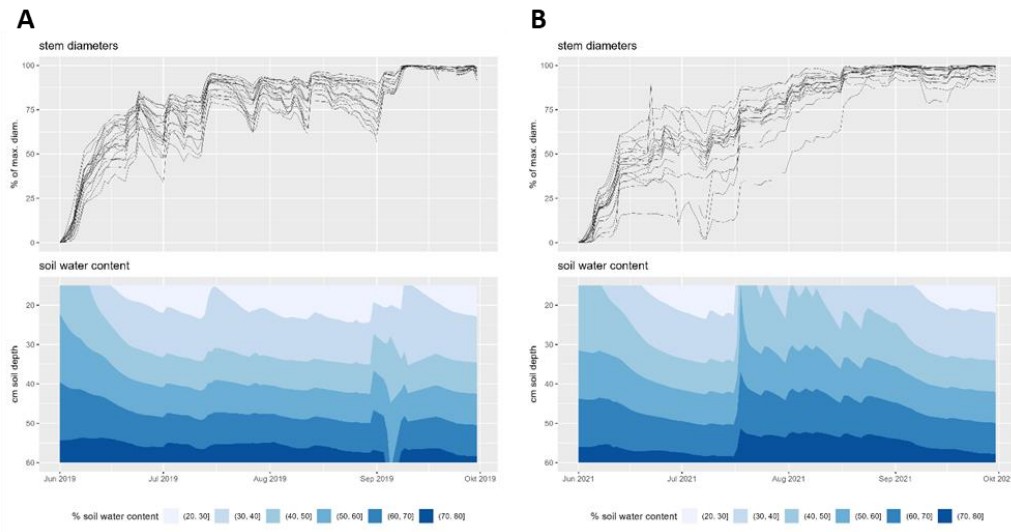

*Figure 5.* Relative stem diameters and soil moisture at the site Klausen-Leopoldsdorf (KLL) during the dry years 2019 (A) and 2021 (B). Stem diameter values were scaled to an annual amplitude of 100.

## 6. Discussion

We provide baseline ecosystem data related to the carbon cycle and capture naturally occurring ECEs across various ecosystem types typical for Austria and other regions of Central Europe. Such data sets are scarce because the measurements are demanding in terms of maintenance and funding. Automated soil respiration data in high temporal resolution, as we report it here, is rare too owing to a lack of dedicated monitoring or research infrastructures (Bond-Lamberty et al., 2021). However, soil $CO_2$ respiration constitutes the second-largest flux in the global carbon cycle, hence is key in estimating ecosystem response to ECEs (Bond-Lamberty and Thomson, 2010). In addition, we provide soil temperature and moisture measurements in the same resolution, being key variables determining soil respiration (Pumpanen et al., 2015). High-resolution measurements of tree stem circumference have been developed as complementary data to relate drought stress with changes in carbon allocation in trees (Zweifel, 2016; Zweifel et al., 2021). The microclimatic, soil, and tree physiological data is complemented by $CO_2$ and $CH_4$ fluxes between the vegetation and the atmosphere measured with Eddy covariance techniques of the two wetland sites.

Our data is particularly useful for drought-related research. Triggered by the pan-European drought of 2003 (Ciais et al., 2005), a key scientific question has been how droughts affect greenhouse gas sinks and sources in ecosystems (Rödenbeck et al., 2020; Reichstein et al., 2013; Anderegg et al., 2020). Droughts usually reduce soil respiration due to the decrease in autotrophic respiration but also because soil microbial activity drops due to water limitation (Grünzweig et al., 2022). Furthermore, rewetting can result in pulses of high soil respiration (Borken and Matzner, 2009). Drought effects on the ecosystem C cycle can persist for years (Kannenberg et al., 2020; Müller and Bahn, 2022) and novel approaches are being developed for assimilating high-resolution data for understanding and quantifying such legacies (Yu et al., 2022; Fu et al., 2020). In this context, the availability of long-term, high-resolution measurements of key ecosystem parameters is key for understanding and quantifying the effects of recurrent droughts (Oberleitner et al., 2022).






The sites presented here are currently being upgraded towards their implementation in the
European Research Infrastructure for Integrated European Long-Term Ecosystem, critical zone and
socio-ecological Research (eLTER RI), together with another ~200 sites in Europe (Mirtl et al., 2018).
Climate change impacts on ecosystem processes including the carbon cycle are among the targeted
research areas the eLTER RI will focus on. The measurements resulting in the data presented here
will continue in future under the umbrella of eLTER RI. Compiling longer-term data series depends
upon the availability of already validated data sets - as it is presented here - before the RI is being
operational. Furthermore, long-term ecosystem observations already exist in these sites with regard
to water and nitrogen cycle allowing for a contextual interpretation of the trends seen in C related
parameters.
Combining several research and monitoring activities at already heavily instrumented sites not only
saves money but widens the data analyses portfolio (Futter et al. 2023; Kulmala 2018). Even though
we provide Eddy covariance data for two of our sites, Austria is not part of the International Carbon
Observation System (ICOS). A combination of data capturing long-term boundary layer exchange of C
together with soil C fluxes, microclimate, and, in forests, tree physiological data obviously holds great
potential (Zweifel et al., 2023; Ramonet et al., 2020). Hence, using the sites simultaneously for other
research infrastructures, such as ICOS, providing high-quality Eddy covariance measurements would
obviously be ideal. The more so because European Research Infrastructures follow the FAIR data
principles to make data Findable, Accessible, Interoperable and Reusable (Wilkinson et al., 2016).
While the eLTER RI data infrastructure is still under development, we comply with the standards
already implemented. We used DEIMS-SDR (https://deims.org/) as the catalogue documenting the
sites (Wohner et al., 2019; Wohner et al., 2022). It issues persistent identifiers for sites (see Table 1)
that allow to uniquely identify sites across research projects and networks. Tools are being
developed to query available information about sites programmatically (Oggioni et al., 2023;
Wohner, 2023) providing contextual ecosystem information.

7.   Data availability
7.1 Data access
The data and metadata is accessible at B2SHARE (https://b2share.eudat.eu/), a service provided by
the EUDAT Collaborative Data Infrastructure. DOIs of the datasets are listed in Table 3. The site
metadata in DEIMS-SDR (Table 1) is part of the data metadata so that site information can easily be
accessed. In chapter 3, we provide a jupyter notebook to download and merge the single datasets,
and to visualize parameters.
*Table 3. Dataset DOIs*

| Site | Dataset | DOI | Reference |
|------|---------|-----|-----------|
| Klausen-Leopoldsdorf | Meteorology | https://doi.org/10.23728/b2share.8f872a3 7513c4768b16ce755eca4bb57 | (Gartner et al., 2024a) |
| | Soil climate | https://doi.org/10.23728/b2share.8d49c0b 557f1455a9e66689e035b8cce | (Gartner et al., 2024b) |
| | Soil $CO_2$ respiration | https://doi.org/10.23728/b2share.5286bd 1bc6aa491f874b9bb12d1c5673 | (Kitzler and Hofbauer, 2024) |
| | Stem increment | https://doi.org/10.23728/b2share.68d84a9 13f0c4875be5c680ad4d6959e | (Gartner and Gollobich, 2024) |





| | | | |
|---|---|---|---|
| Rosalia Forest Demonstration Centre | Meteorology | https://doi.org/10.23728/b2share.96c52c247eb846deb2a3ec5e2c27b4f1 | (Diaz-Pines, 2024a) |
| | Soil climate | https://doi.org/10.23728/b2share.c68143fc11224c44ae5529bd6a35a76d | (Diaz-Pines, 2024c) |
| | Soil CO$_2$ respiration | https://doi.org/10.23728/b2share.d167e727abe947abbc8efc04057557f6 | (Diaz-Pines, 2024b) |
| | Stem increment | https://doi.org/10.23728/b2share.d0d185f1eb184ae48f6d06ea9aa8dbdf | (Diaz-Pines, 2024d) |
| Zöbelboden | Meteorology | https://doi.org/10.23728/b2share.762e665273234b129d09ef017416bcfb | (Kobler et al., 2024a) |
| | Soil climate | https://doi.org/10.23728/b2share.46e19191ce9c427d90f48ce38f56a0e1 | (Kobler et al., 2024c) |
| | Soil CO$_2$ respiration | https://doi.org/10.23728/b2share.4f44006b932142e68981106a016f1f56 | (Kobler et al., 2024b) |
| | Stem increment | https://doi.org/10.23728/b2share.2de5b37a0cad4f82a19f477531d6af24 | (Pröll et al., 2024) |
| Stubai - Kaserstattalm | Meteorology | https://doi.org/10.23728/b2share.77462914dc0b43cb8c24a967e6851665 | (Ingrisch and Bahn, 2024c) |
| | Soil climate | https://doi.org/10.23728/b2share.026d76094e8f4512b09b35b7a0d2a9d7 | (Ingrisch and Bahn, 2024d) |
| | Soil CO$_2$ respiration | https://doi.org/10.23728/b2share.cfe8c7ad1965433484650ea9026512ca | (Ingrisch and Bahn, 2024a) |
| | Stem increment | https://doi.org/10.23728/b2share.0e3eed54ff30418f8720806b5f05cca9 | (Ingrisch and Bahn, 2024b) |
| Pürgschachen Moor | Meteorology | https://doi.org/10.23728/b2share.5442510ad03e4968afb4e2108e85a64d | (Maier and Glatzel, 2024e) |
| | Soil climate | https://doi.org/10.23728/b2share.9380364098d14978b876a87517652d62 | (Maier and Glatzel, 2024f) |
| | Eddy Covariance | https://doi.org/10.23728/b2share.4f783e3ff2884abca5c59960db0b7955 | (Maier and Glatzel, 2024d) |
| Lake Neusiedl | Meteorology | https://doi.org/10.23728/b2share.f7176c9ee982464f947d2fe9fb8f389d | (Maier and Glatzel, 2024b) |
| | Soil climate | https://doi.org/10.23728/b2share.4e6474cd55f9487d97e3d31e83baa530 | (Maier and Glatzel, 2024c) |
| | Eddy Covariance | https://doi.org/10.23728/b2share.b83caca3efe44868a1ed49129b4a576a | (Maier and Glatzel, 2024a) |


7.2 Data visualization, workflow integration
The software stack used to store, import and quality control the provided data is built on PostgreSQL
database with a Post-GIS extension. The database structure is derived from the Time Series
Management (TSM) system developed by the Research Center Jülich (Wohner, C., Dirnböck, T.,
Peterseil, J., Pröll, G., Geiger, S., 2021) and originally deployed during the LTER CWN project but was
repurposed to better fit the needs of the data management and working group. Now, for the import
and quality control of data, a number of Python scripts deployed in a Jupyter environment are used.
This is also includes scripts to visualise the data on the fly in Jupyter.

8.  Code availability

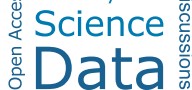

A Jupyter notebook to access, merge, and visualize the data from all sites is available at
https://gist.github.com/1O/9bbe44a03f12801c6c742202b005db57.

9.  Author contribution
DT, BM, DPM, EM, GK, GG, HA, IJ, KB, KJ, MA, PG, VS, ZA, and GS designed the measurements and
carried them out. WC, PJ designed and constructed the database. KK, VS, and PG customized and
filled the database. OI developed the Jupyter notebook. DT prepared the manuscript with
contributions from all co-authors.

10. Competing interests
The authors declare that they have no conflict of interest.

11. Acknowledgements
We want to thank Manfred Bogner, Thomas Lehner, Christian Holtermann, Thomas Kager, and Josef
Gasch for technical implementation and assistance.

12. Funding
The infrastructure and its implementation was funded by the Austrian Research Promotion Agency
(FFG, project LTER-CWN: Long-Term Ecosystem Research Infrastructure for Carbon, Water and
Nitrogen, grant no. 858024). The Austrian Academy of Sciences (ÖAW) supported all authors for data
compilation and writing of the manuscript through its eLTER 2022 call (Earth System Sciences (ESS)).
T.D., J.K., K.K., J.P., C.W. and E.D-P. received additional funding from the EU Horizon 2020 project
eLTER PLUS (grant no. 871128), and E.D-P. also from the project EXAFOR (Austrian Climate Research
Programme 12th Call, grant no. KR19AC0K17557).

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
