# Peer review of "High-resolution Carbon cycling data from 2019 to 2021 measured at six"

_Earth System Science Data, 2024_

## Author Response (AR1)

**Dear Editor,**

We herewith submit the revised version of our manuscript "High-resolution Carbon cycling data from 2019 to 2021 measured at six Austrian Long-Term Ecosystem Research sites". In addition to the changes made in responses to two referees comments (see below), we made the following minor change.

We changed the names of two co-authors: Hofbauer to Malli since the colleague changed name meanwhile; Zechmeister to Zechmeister-Boltenstern since we missed the double name when submitting

We further want to make the following suggestion regarding Table 1. Both referees suggested more detailed information in this table: Metadata Table 1: since very detailed description of all sites is available via the cited link to DEIMS-SDR system, we do not want to overload the table and just added the most important information in order to more easily comprehend the data graphs in the manuscript.

We want to emphasize that the comments were very helpful to improve our work.

We hope that the revised version made the necessary adaptations to our initial submission so that it is now suitable for publication.

With best wishes

Thomas Dirnböck (on behalf of all authors)

Vienna, 2024-08-28

**Response to Reviewer #1**

**Overview and general recommendation:**

Extreme events are projected to happen more frequently in the future under the ongoing climate change. Understanding how ecosystem carbon fluxes and their components (e.g. soil respiration) respond to these extreme events, and identifying the critical regulators of these responses, is crucial. High-quality datasets are essential to address these questions. Despite a slightly short coverage of years, the carbon fluxes and relevant environmental factors at six Austrian sites, covering forests, grasslands, and wetlands, provided by Dirnböck et al., are valuable for quantifying the effects of extreme events on carbon dynamics. Overall, the manuscript clearly describes the site information, instrument setup, available data at each site, and relevant data processing procedures. I believe it can be a valuable dataset, but some issues need to be addressed before the manuscript is suitable for publication.

*Author response: we are grateful for the positive evaluation and provide a point-by-point response to the major and minor comments below*

**Major Comments:**

1. As one of the most important carbon cycling data, the ecosystem-level eddy covariance data have not been displayed in this manuscript. We suggested authors to display the EC data at least for one site in each vegetation type.

*Author response: thank you for the suggestion. We will add this Fig. to the manuscript. Kindly note that our dataset includes EC data from two wetland sites (see table 3 of the manuscript); please see the figure for the site PUE as an example.*

2. It is interesting and useful to have the continuous soil respiration observations in some of sites. However, there are lots of missing data especially in critical period in the growing season. I am wondering if this dataset can really achieve the goal to understand the effects of extreme events on carbon dynamics.

*Author response: we agree that even three years of this kind of data can only to some extent be used to explore effects of climate extreme events. Therefore, we pointed out that "We provide baseline ecosystem data related to the carbon cycle and capture naturally occurring ECEs across various ecosystem types typical for Austria and other regions of Central Europe." (Discussion L 350). As a baseline, the data covers a first set of three years, which will be prolonged in the Ecosystem Research Infrastructure (eLTER RI) and is supplemented with historical data published in previous manuscripts and data repositories (see Discussion in L379-L384). It is clear and unfortunate that the dataset has some data gaps (usually due to the malfunctioning of devices and complications during the installation and adjustment phases). In order to strengthen these limitations,*

we added *"While the three-year data with the usual measurement gaps can only to some extent capture aspects of drought related effects it represents a valuable baseline." (L373-L375).*

3. A lot of abbreviations in the manuscript have not been clearly defined. To reach wider audiences and fully exploit the data, better definition of terms is needed.

*Author response: we thank the reviewer for spotting this issue. We will add all the details for the abbreviations (see detailed response below)*

Additionally, even though the manuscript is generally clearly described, text can be polished to make the description more concise and well linked.

*Author response: we will rework the entire text.*

**Minor Comments:**

1. Line 24: What "the high resolution" is? Half-hourly or daily? Could you specify them?

*Author response: Here, we will specify the range "(15 – 60 minutes)", and will add the specific resolution to each measurement component*

2. Line 55: "C" has not been defined before.

*Author response: Right indeed, we will write C (carbon) in L34*

3. Line 52: From the abbreviation of "LTER-CWN", the description before should be "carbon, water, and nitrogen"?

*Author response: Thanks for pointing to this. We will add the long title.*

4. Line 66-69: The names are not exactly consistent with the descriptions in Figure 1. Where is Kaserstattalm? In the site descriptions, authors also mentioned lowlands et al., it could be easier to understand if the authors could also add elevation in the map.

*Author response: Thanks again. We will change the name to "Stubai", which is the site name and Kasterstattalm is a Subsite*

5. Line 74: What is the "DEIMS-SDR" stands for, can you spell its full name?

*Author response:* Thanks, we again will spell out the abbreviation.

6. Line 76: Please specify the full name of "BOKU", this also applies for "FAO" and "WRB" in Line 83.

*Author response: Since 2024 it is recommended to refer to BOKU as "BOKU University" in English; the alternative would be to use the official name Universität für Bodenkultur, which can be confusing for an international audience. We will added the correct reference for FAO/WRB at all placed it was cited.*

7. Line 91: What is the meaning of "DRAIN" and where its location in the map?

*Author response: Instead of adding the full name, we will skipp the abbreviation, as it is not considered crucial.*

8. Line 94: What the selected soil biogeochemical and microbiological processes?

*Author response: We suggest to keep this general description, since it is not considered important for the data we present here.*

9. Line 104: Please specify the full name of "ICP" and explain what is the Level 2 site.

*Author response: Again, we will spell out the abbreviation for ICP Forests, but will skipp "Level 2" because it is not important in the context of the manuscript.*

10. Line 176: What "UNECE" stands for?

*Author response: We will change the abbreviation to "International Cooperative Programme on Integrated Monitoring of Air Pollution Effects on Ecosystems (ICP IM)".*

11. Line 198: The vertical line of "Zöbelboden" is not visible.

*Author response: Thanks for pointing this out. We will change the figure accordingly.*

12. Line 199: Please properly define the LTSER.

*Author response: We will do so by writing "Long-term socio-ecological research platforms (LTSER)".*

13. Line 201-202: For Metadata Table 1, maybe the authors can add the information such as elevation, annual temperature and precipitation, dominant species, main equipped measurements, "data shared status" et al., to make the table more comprehensive so that the audience can have a general overview of the introduced Austrian LTER sites.

*Author response: We will add the requested metadata after clarifying with the editor how comprehensive the overall table can be. In any case, we consider the DEIMS-SDR link - being rather long - crucial because this is where the site metadata is located.*

14. Line 264-265: To confirm, the "Eddy covariance devices" were calibrated once a year or monthly?

*Author response: In order to clarify this, we will write: "calibrated once a year in the lab, and monthly in the field."*

15. Line 274: What is the time resolution for the measurements of stem growth?

*Author response: We added the time resolutions of all measurement components.*

16. Line 294-295: "At KAS, the maximum temperatures in the year 2021 were lower (0.6 °C)"—compared to when?

*Author response: Thanks for the question; this part was indeed misleading. We now write: "The mean annual temperature maxima (90 percentile) were between 0.3 °C (KAS) and 2.3 °C (ZOE) higher in the year 2019 than in 2020. These differences were lower when comparing in the year 2021 with 2019 (≤< 0.5 6 °C)."*

17. Line 297: It seems that Figure 3 is not about the soil temperature.

*Author response: We apologize; the correct figure is Figure 4.*

18. Line 299: In Figure 2, you may add the long-term monthly mean temperature and precipitation to better illustrate the 2019 and 2021 are drier years while 2020 is roughly close to average climate?

*Author response: Indeed, we used the year 2020 as a reference for an average climate (see L297-L298). After discussing ways to show the long-term averages we suggest to use gridded long-term data from the Austrian meteorological service ([https://data.hub.geosphere.at/dataset/winfore-v2-1d-1km](https://data.hub.geosphere.at/dataset/winfore-v2-1d-1km)). With this data, we are able to show - consistently for all sites - deviations of a drought index (SPEI - Standardized Precipitation Evapotranspiration Index) for 2019, 2020, and 2021 from a 30-year average (1980-2010). Using this new analysis, we will add to the chapter and adapt in the following way: "We used gridded SPEI (Standardized Precipitation Evapotranspiration Index) from the Austrian Meteorological Service (https://data.hub.geosphere.at/dataset/winfore-v2-1d-1km; Haslinger & Bartsch (2016)) to compare the long-term average water availability during the growing season (1980-2010; May to September) with those occurring in the measurement years (Table 3). The advantage of the SPEI is that it accounts for precipitation and temperature via evapotranspiration and integrates over a given temporal window (we used 30 days) (Vicente-Serrano et al. 2010) Accordingly, the 2021 was closest to the long-term average, the year 2020 was a particularly wet year, and the year 2019 was drier than the average. However, there were differences between the sites: Particularly the mountain station in the Tyrolian Alps (KAS) did not experience significant deviations in SPEI as compared to the long-term average apart from a wet growing season in 2021. The SPEI at the site in the Viennese Forest (KLL) does not indicate that in 2019, the growth period was*

*particularly dry. The monthly precipitation and temperature patterns are shown in Figure 2, and soil water content and soil temperatures in Figure 3 and Figure 5. Differences in the seasonal precipitation patterns between the measurement years vary a lot between sites. In sum, lower precipitation occurred in 2019 and 2021 than in 2020 in all sites. The mean annual temperature maxima (90 percentile) were between 0.3 °C (KAS) and 2.3 °C (ZOE) higher in the year 2019 than in 2020. These differences were lower when comparing the year 2021 with 2019 (≤ 0.6 °C). In accordance with SPEI, precipitation and temperature, soil water content showed the lowest values during the years 2019 followed by the year 2021, and soil temperature were higher during these years (Figure 4)." (please see the SPEI results in the attachment; the table will be included in the manuscript)*

19. Line 314: Rs should be defined

*Author response: We will do so: "Soil respiration (Rs)".*

20. Line 335-337: How do you conclude this, do you have some analysis to demonstrate this statement?

*Author response: Thank you for your question. We agree that it is not possible to draw such a conclusion from the way we present the data in this paper. While we don't think that such a specific analysis would be needed for a data paper, we could provide if the editor considers this to be helpful or needed. Our suggestion is to included some references, which back up the statement for the sites.*

21. Line 345: It seems that there are quite large variations in stem growth between 2019 and 2021, how was the stem growth in normal year 2020?

*Author response: We suggest adding the data from the year 2020 in a then three-panel figure. In 2020, higher soil moisture particularly during spring and early summer (the main growth period) leads to a continuously increasing stem diameter without the pumps occurring during dry periods in 2019 and 2021 (we add this figure as an attachment).*

22. Line 387: ICOS should be defined as "Integrated Carbon Observation System".

*Author response: We will changed the text accordingly.*

**Response to Reviewer #2**

**Summary**

This study provided observational data from seven sites, including meteorological and soil variables, CO2 fluxes, and tree stem growth during 2019-2021. These observations are essential for understanding and modeling of ecosystem carbon responses to climate variations or extreme climate events. The authors provided detailed site descriptions and how they measured the data and conducted the QA/QC. They also visualized the data and briefly analyzed the data. The manuscript is generally well organized. However, to further improve the quality of this paper and increase the data's impacts, please address my comments probably below.

*Author response: thank you for the positive feedback and evaluation. Please find a point-by-point response to your comments below*

1)   Title: 'LTER', please give its full name since many readers might not know what does such an acronym mean.

*Author response: we agree and will provide the full name for the abbreviation.*

2)   Abstract, line 19: seven 'long-term' observation sites. I understand that the seven sites belong to the LTER and will continue providing new data when available in the future. However, ESSD is a data journal, and this paper only provided data during 2019-2021, which are not long-term observations. So it's a bit tricky to refer to these sites as long-term sites now. If so, I would say lots of sites are long-term sites since they will also provide measures in the future, however, this may mislead data-users.

*Author response: we understand your point when only considering the carbon-related flux variables. However, the sites are indeed long-term observation sites because they already exist for many years providing long-term data for numerous ecosystem variables, including some of those compiled in this manuscript for the years 2019-2021 (e.g. meteorology, soil temperature and moisture). We would therefore like to keep the term "long-term".*

3)   Line 45: add comma behind 'ecosystem carbon cycling'

*Author response: we will do so*

4)  Line 55: for the high temporal resolution, what's the temporal resolution for each site?

*Author response: this detail was also asked for by reviewer #1. We will specify the specific resolution to each measurement component for each site in chapter 3.*

5)  Table 1 is important for readers to quickly learn about the sites. However, this table only provided very limited information. Although there are some descriptions for each site in the main text, it would be great to explicitly list the key information for each site in Table 1, such as the annual mean meteorological conditions, latitude and longitude, major land/vegetation covers, CO2 flux measurement period (which year and month is measured vs. what period is not covered), what environmental variables are measured for each site.

*Author response: reviewer #1 asked for such information as well. Therefore, we will add the requested metadata after clarifying with the editor how comprehensive the overall table can be. In any case, we consider the DEIMS-SDR link - being rather long - crucial because this is where the detailed site metadata is located.*

6)  Line 217: relative or specific humidity?

*Author response: it is relative humidity; we will add this specification*

7)  Line 217-218: 'several radiation variables', please clearly list what variables are measured for each site so that the readers can easily understand the data availability for each site.

*Author response: we agree and will add an additional table specifying the meteorological data*

8)  This study mentions multiple times of CH4 fluxes measured. Do the authors also intend to make the CH4 fluxes publicly available in this study?

*Author response: yes, CO4 eddy covariance data is provided by the two wetland sites (Pürschachen Moor bog and the reed belt of lake Neusiedler See). Please see chapter 3.2.2. In response to a suggestion of reviewer #2, we will add an additional Figure visualizing this data (see uploaded Figure "eddy_flux_pue.png").*

9)   Line 263-265: what do you mean 'monthly in the field'? checked or calibrated?

*Author response: we will specify this by writing "The Eddy Covariance devices were checked daily via remote access, calibrated once a year in the lab, and monthly in the field"*

10)   Line 272, why data-set without micro-meteorological conditions was regarded as gaps?

*Author response: Thank you for making us aware. Here, we indeed described it the wrong way because the quality flags (0-2) obviously result from the use of the micrometeorological Eddy Covariance data, too. Therefore we will change the sentence to: "In addition, gaps resulted from power breakdowns".*

11)   Line 288-297, this paragraph is quite confusing. Please revise it carefully. For example, line 288-289: what do you mean 'long-term averages' here? How did you draw the conclusion that 2020 is closer to average while 2019/2021 drier/warmer? It's unclear from Fig. 2.

*Author response: we agree, this is a weak part of the manuscript. Reviewer #1 spotted it too (see comment 18). The assessment of the meteorological conditions of the three years were now done in a different way and the entire paragraph will be rewritten in the following way: "We used gridded SPEI (Standardized Precipitation Evapotranspiration Index) from the Austrian Meteorological Service (https://data.hub.geosphere.at/dataset/winfore-v2-1d-1km; Haslinger & Bartsch (2016)) to compare the long-term average water availability during the growing season (1980-2010; May to September) with those occurring in the measurement years (Table 3). The advantage of the SPEI is that it accounts for precipitation and temperature via evapotranspiration and integrates over a given temporal window (we used 30 days) (Vicente-Serrano et al. 2010) Accordingly, the 2021 was closest to the long-term average, the year 2020 was a particularly wet year, and the year 2019 was drier than the average. However, there were differences between the sites: Particularly the mountain station in the Tyrolian Alps (KAS) did not experience significant deviations in SPEI as compared to the long-term average apart from a wet growing season in 2021. The SPEI at the site in the Viennese Forest (KLL) does not indicate that in 2019, the growth period was particularly dry. The monthly precipitation and temperature patterns are shown in Figure 2, and soil water content and soil temperatures in Figure 3 and Figure 5. Differences in the seasonal precipitation patterns between the measurement years vary a lot between sites. In sum, lower precipitation occurred in 2019 and 2021 than in 2020 in all sites. The mean annual temperature maxima (90 percentile) were between 0.3 °C (KAS) and 2.3 °C (ZOE) higher in the year 2019 than in 2020. These differences were lower when comparing the year 2021 with 2019 (≤ 0.6 °C). In accordance with SPEI, precipitation and*

*temperature, soil water content showed the lowest values during the years 2019 followed by the year 2021, and soil temperature were higher during these years (Figure 4)." The new Table 3 was uploaded.*

12)    Line 291-292, 'In sum, the dry periods resulted in lower precipitation in 2019 and 2021 in all sites.' It's really hard to directly retrieve this information from Fig.2. Please use some statistical number to support this.

*Author response: please see the response to comment 11 above*

13)    Line 292-295: the same issue. Hard to understand.

*Author response: please see the response to comment 11 above*

14)    Line 295-297: you mean Fig.4? Honestly, it's hard to draw such a conclusion directly from Fig.4. You mean annual mean soil water content/temperature here?

*Author response: please see the response to comment 11 above*

15)    Line 304: what's the snow-free period? Please also see my comment#5.

*Author response: we agree that this is important information and will add it to Table 1 (see our response to comment 5 above)*

16)    Line 318-320: what kinds of statistics test did you use?

*Author response: we used a t-test statistic. We will add this to the p-values*

17)    Line 325-326: could you please clearly show the magnitude of the spatial variation in CO2 fluxes and the magnitude difference caused by measurement devices, and then get such a conclusion. The current statement always makes me feel less rigorous.

*Author response: We are not entirely sure whether we understand the comment. Figure 3 visualizes spatial variation (error bars) and uncertainty in measurements (at least a proxy*

*since we can only compare automated and manual chambers). The latter is shown in the deviation of the two regression lines in each plot. The conclusion we derive is taken from Figure 3. We will add an additional reference to Figure 3 after the conclusion in order to make this clear.*

18)   Line 335-337: again, it's hard to see such conclusion directly from Fig. 4.

*Author response: Reviewer #1 made us aware of this issue, too. We agree that it is not possible to draw such a conclusion from the way we present the data in this paper. While we don't think that such a specific analysis would be needed for a data paper, we could provide if the editor considers this to be helpful or needed. Our suggestion is to included some references, which back up the statement for the sites.*

19)   Fig. 4, what's the temporal resolution for each variable? From Fig. 4, it seems that the temperature and water content are continually measured. Why not aggregate to the same temporal resolution to soil respiration and then show the data?

*Author response: We have again discussed how to visualize the data and see no particular advantage in aggregating the CO2 data. One drawback among others is that we would lose temporal coverage in our visualization. We suggest to keep Figure 4 as it is.*

20)   Why not include CH4 flux in the results?

*Author response: we refer to our response to comment 8.*

---

## Referee Report (RR1)

Generally, the authors have made some revisions and improvements to the manuscript according to the reviewers' comments. However, I feel the authors did not try their best to improve the manuscript. There are still the following points that the authors need to pay attention:

1)      In the "response to reviewers", authors often did not mention the lines where they have made the revisions, which made the review process inconvenient. In some cases, the authors also mentioned wrongly for the line numbers (e.g. Line numbers for the response to the major comment 2 from reviewer 1) – I feel the authors could take the revision more seriously.

2)      Sometimes the authors did not respond directly or ignore the comments from authors (e.g. to indicate the elevation in the map) – It is definitely fair and reasonable to not accept all the comments from the reviewers. But it is good to give a reasonable reasoning, and in most of cases, I think it only needs at most 1-2 phrases or sentences to explain the situation (e.g. comment 8 for reviewer 1). Besides, sometimes the authors mention they would not change the phrasing but to add some references to support their statement, but they did not specify what they have added (e.g. comment 20 from reviewer 1).

3)      Sometimes, the authors could think more to make the information more efficient—for instance, for Table 1, both reviewers 1 and 2 suggested adding some more details in the Table to make the audience understand easily. Of course, as the authors indicated, we can find all the information on the website link – but I think you would like to make your data be used by peers as often as possible, not just publish a paper, right? Anyway, the weblink is long, but I think you can use for instance some hyperlinks to avoid the long characters.

4)      The mean SPEI between 1980-2010 should not be useful enough as a reference to compare with the SPEI data between 2019-2021. As the SPEI is a normalized index considering the distribution of water deficit in the study period (1980-2010), that's why the authors can notice the mean of SPEI between 1980-2010 is close to 0. Hence, in this situation, the mean SPEI between 1980-2010 is not informative and wrongly used. Somehow, it could be rather simple to use P-PET as an indicator to illustrate the water status in 2019-2020 compared to the historical period in each site.

5)      The figures that reviewers provided sometimes look vague – I suggest the authors enlarge the font size to make the figures more readable (if possible, to beautify some plots). Besides, all the figures should be clearly described in the figure caption.

As several senior and even well-established scientists on the coauthor list, I feel the manuscript should be better presented compared to the current version.

---

## Author Response (AR2)

**Dear Editor,**

please find the attached revised version of our manuscript "High-resolution Carbon cycling data from 2019 to 2021 measured at six Austrian Long-Term Ecosystem Research sites".

We would like to express our sincere thanks for the work the reviewer has done. While we believe that the criticism is not justified everywhere, we also think that this may have been partly due to some misunderstandings. Please see below our point-by-point responses to the reviewer comments and the additional changes we have made. We hope that the revised version entails the necessary adaptations to our last submission so that it is now suitable for publication.

Sincerely

Thomas Dirnböck (on behalf of all authors)

Vienna, 2024-12-02

**Response to Reviewer**

**Overview and general recommendation:**

Generally, the authors have made some revisions and improvements to the manuscript according to the reviewers' comments. However, I feel the authors did not try their best to improve the manuscript. There are still the following points that the authors need to pay attention:

1) In the "response to reviewers", authors often did not mention the lines where they have made the revisions, which made the review process inconvenient. In some cases, the authors also mentioned wrongly for the line numbers (e.g. Line numbers for the response to the major comment 2 from reviewer 1) I feel the authors could take the revision more seriously.

*Response: We are sorry that the reviewer had the impression that we did not take the revision seriously, as taken from a lack of line numbers in our previous responses. We should point out that it was not possible to provide the line numbers of the revised manuscript, given that ESSD requires uploading all responses before providing the authorization to submit a revised version of the manuscript. Thus, the revised version of the manuscript did not yet exist when the responses were elaborated and, therefore, respective line numbers of the new version could not be added. To overcome this issue, the latest point by point response to both reviews included a throughout description of the intended changes, and, when submitting the new version of the manuscript, we uploaded a track-changed version of the manuscript wherein all the detailed changes were marked up.*

2) Sometimes the authors did not respond directly or ignore the comments from authors (e.g. to indicate the elevation in the map) It is definitely fair and reasonable to not accept all the comments from the reviewers. But it is good to give a reasonable reasoning, and in most of cases, I think it only needs at most 1 2 phrases or sentences to explain the situation (e.g. comment 8 for reviewer 1). Besides, sometimes the authors mention they would not change the phrasing but to add some references to support their statement, but they did not specify what they have added (e.g. comment 20 from reviewer 1).

*Response: We would like to once again express our gratitude to both reviewers, who provided very useful suggestions to improve our work. In almost all cases, we adapted and added information in accordance with these suggestions and described in detail what we did and why. In the few cases where we did not, we provided a justification. Only in the case of the suggestion to add elevation to the map, we did not.*

*Reviewer 1 suggested to add elevation in the map (figure 1). We apologize for not responding to this suggestion. The reason why we decided not to add this information to the map was that we wanted to keep the map simple and graphically appealing. Additional information on the sites, including altitude and climate, were included in Table 1. To address the reviewers comment and make this more obvious, we have now added the following sentence to the caption of Fig. 1: "For site information concerning ecosystem type, altitude and climate see Table 1", and have also modified the caption of Table 1 to make clear that it also contains this information. In addition, we slightly adapted the presentation of Table 1.*

*Comment 8 of Reviewer 1: In the new version, we specified the sentence to "This experiment focuses on investigating the effect of changing precipitation patterns on soil nitrogen fluxes, soil*

microbial changes, greenhouse gas efflux, and soil water processes." *(L95-97, track-change version)*

*Comment 20 of Reviewer 1: In the new version, we added a reference from a study about the causes of spatial soil respiration variation at the site ZOE, which supports the statement (L355, track-change version).*

3) Sometimes, the authors could think more to make the information more efficient for instance, for Table 1, both reviewers 1 and 2 suggested adding some more details in the Table to make the audience understand easily. Of course, as the authors indicated, we can find all the information on the website link but I think you would like to make your data be used by peers as often as possible, not just publish a paper, right? Anyway, the weblink is long, but I think you can use for instance some hyperlinks to avoid the long characters.

*Response: In our previous response to the reviewers we point out that "Both referees suggested more detailed information in this table: Metadata Table 1: since very detailed description of all sites is available via the cited link to DEIMS-SDR system, we do not want to overload the table and just added the most important information..." The additional information we included is ecosystem type, altitude of the sites, annual mean temperature, and mean annual precipitation.*

*We thank the reviewer for suggesting the use of hyperlinks to avoid the rather lengthy links to the DEIMS-SDR ID. We will discuss with the Journal editorial office whether this is possible*

4) The mean SPEI between 1980 2010 should not be useful enough as a reference to compare with the SPEI data between 2019 2021. As the SPEI is a normalized index considering the distribution of water deficit in the study period (1980 2010), that's why the authors can notice the mean of SPEI between 1980 2010 is close to 0. Hence, in this situation, the mean SPEI between 1980 2010 is not informative and wrongly used. Somehow, it could be rather simple to use P PET as an indicator to illustrate the water status in 2019 2020 compared to the historical period in each site.

*Response: We did not calculate SPEI for different periods but used existing gridded data, which was calculated for the period between 1960 and 2021 (and normalized over this period) and extracted the SPEI values for the sites for each year. Then, we compared the mean SPEI values for the measurement years (2019 to 2021) with the 30-year period before these years (1980 to 2010). The SPEI between 1980 to 2010 is close to zero because it is close to the average drought water balance between 1960 and 2021 (only slightly drier). Hence, according to our understanding, the usage of SPEI was correct, and conclusively indicates the drought situation at each site in each of the measurement years compared to the long-term average. In order to avoid any misunderstanding, we added: "Note, that gridded SPEI data set is based on meteorological data for the period 1960 to 2021" to the respective section in L313-L314 (track change version).*

5) The figures that reviewers provided sometimes look vague I suggest the authors enlarge the font size to make the figures more readable (if possible, to beautify some plots). Besides, all the figures should be clearly described in the figure caption.

*Response: we agree that the font sizes of Figure 4, 5, and 6 are too small. We changed them and accommodated the axes information. And, we adapted the captions accordingly.*

**Additional changes**

Since one of the authors changed his family after submitting, we changed the name and reordered the authors alphabetically (besides first and last author).